# Effective Neural Network $L_0$ Regularization With BinMask

## Abstract

$L_0$ regularization of neural networks is a fundamental problem. In addition to regularizing models for better generalizability, $L_0$ regularization also applies to selecting input features and training sparse neural networks. There is a large body of research on related topics, some with quite complicated methods. In this paper, we show that a straightforward formulation, BinMask, which multiplies weights with deterministic binary masks and uses the identity straight-through estimator for backpropagation, is an effective $L_0$ regularizer. We evaluate BinMask on three tasks: feature selection, network sparsification, and model regularization. Despite its simplicity, BinMask achieves competitive performance on all the benchmarks without task-specific tuning compared to methods designed for each task. Our results suggest that decoupling weights from mask optimization, which has been widely adopted by previous work, is a key component for effective $L_0$ regularization.

## 1 Introduction

Regularization is a long-standing topic in machine learning. While $L_1$ and $L_2$ regularization have received thorough theoretical treatment and been widely applied in machine learning, $L_0$ regularization is still under active investigation due to the optimization challenges caused by its combinatorial nature. Given a vector $\boldsymbol{W} \in \mathbb{R}^n$, $L_0$ regularization penalizes $\|\boldsymbol{W}\|_0$, the number of non-zero entries in $\boldsymbol{W}$. One prominent application of $L_0$ regularization is to induce sparse neural networks, potentially improving time and energy costs at inference with minimum impact on accuracy (Chen et al., 2020). When applied to the input features, $L_0$ regularization can also select a subset of features important to the learning task.

Increasingly complicated methods have been proposed for $L_0$ regularization of neural networks based on various relaxation formulations (Louizos et al., 2018; Yamada et al., 2020; Zhou et al., 2021). In this paper, we revisit a straightforward formulation, BinMask, that multiplies the weights with deterministic binary masks (Jia & Rinard, 2020). The masks are obtained by quantizing underlying real-valued weights, which are simultaneously updated with network weights during training. BinMask uses the identity straight-through estimator to propagate gradient through the quantization operator by treating it as an identity function during backpropagation. BinMask has demonstrated empirical success in training low-complexity sparse binarized neural networks (Jia & Rinard, 2020).

Unlike most research that only focuses on either feature selection or network sparsification, we evaluate Bin-Mask on three different tasks: feature selection, network sparsification, and model regularization. We use the same BinMask implementation without task-specific hyperparameter tuning. BinMask achieves competitive performance on all three tasks compared to methods designed for each task. Our results suggest that the decoupled optimization of network weights and sparsity masks is a key component for effective $L_0$ regularization. When combined with standard techniques for optimizing binary masks, such a formulation is sufficient to give impressive results. We will also open source our implementation, which supports automatically patching most existing PyTorch networks to use sparse weights with minimum additional code.

## 2 Related work

Xiao et al. (2019) and Jia & Rinard (2020) are most relevant to our work. They both use a similar formulation that multiplies weights with deterministic binary masks optimized via the straight-through estimator.

However, Xiao et al. (2019) uses bi-level optimization with a custom optimizer to update weights and masks separately, which is computationally expensive. Their experiments focus on training sparse networks starting from pretrained networks without evaluating $L_0$ regularization in other scenarios. Jia & Rinard (2020) proposes the name BinMask and uses it to train sparse binarized neural networks with more balanced layer-wise sparsities to achieve tractable verification. However, they do not evaluate BinMask on other applications or real-valued networks. By contrast, our method learns weights and masks simultaneously in each iteration from random initialization, and we evaluate our method on a diverse set of applications.

Another family of closely related methods exploits probabilistic reparameterization and relaxation to the $L_0$ "norm" (Srinivas et al., 2017; Louizos et al., 2018; Zhou et al., 2021). They focus on sparse training and have not evaluated their methods for other applications. Among them, ProbMask (Zhou et al., 2021) achieves the best accuracies on sparse training. However, their method is tailored to network sparsification and is not directly applicable to feature selection or model regularization. A drawback of methods using probabilistic masks is the need to trade off between computation time and estimation variance. Louizos et al. (2018) uses one sample per minibatch despite larger variance. ProbMask uses multiple samples per minibatch at the cost of slower training. Sampling of masks also introduces distribution shift that requires finetuning network weights after fixing masks, as will be shown in our experiments for ProbMask. By contrast, BinMask uses deterministic masks, which is faster to train and can deliver well-performing models without finetuning.

$L_0$ regularization has two critical applications: feature selection and sparse neural network training. There are enormous methods for both tasks that do not formulate them as $L_0$ regularization problems. Feature selection algorithms can be divided into three categories (Tang et al., 2014). *Filter models* select features according to ranks computed by some predefined criterion (Gu et al., 2011; Kira & Rendell, 1992; Vergara & Estévez, 2014). They are computationally efficient but deliver worse results due to the ignorance of the target learner to be used with selected features. *Wrapper models* use heuristics to select a subset of features that maximize the performance of a given black-box learner (El Aboudi & Benhlima, 2016), which is computationally expensive because the target learner needs to be evaluated multiple times. *Embedded models* attempt to address the drawbacks of the other two methods by simultaneously selecting features and solving the learning task (Liu & Yu, 2005; Ma & Huang, 2008). Recently, there has been an increasing interest in using neural network methods for feature selection (Lu et al., 2018; Zhang et al., 2019; Wojtas & Chen, 2020; Yamada et al., 2020; Lemhadri et al., 2021). Some utilize sparse neural networks to derive selected features from the sparsity patterns (Atashgahi et al., 2022; 2023). Yamada et al. (2020) is most relevant to our method, as they multiply the input features of neural networks with stochastic binary gates. However, they preserve continuous gate variables during training, which is different from how selected features are used at test time and thus causes lower performance. Their gate probabilities do not converge in our experiments.

Sparse neural networks promise faster and more energy-efficient inference, although there are software/hardware implementation challenges (Lu et al., 2019). Sparse networks can be obtained with diverse pruning methods that remove neurons from trained dense networks according to different importance criterion (Mozer & Smolensky, 1989; Han et al., 2015; Blalock et al., 2020). Those methods typically need more computation budget because a dense network must be trained before pruning. Another line of work learns weights and sparse connections simultaneously during training, typically via applying learnable masks on the weights or via thresholding the weights (Srinivas et al., 2017; Louizos et al., 2018; Xiao et al., 2019; Jia & Rinard, 2020; Zhou et al., 2021; Vanderschueren & De Vleeschouwer, 2023). It is also possible to train sparse weights from random initialization for certain tasks with proper sparse topology (Frankle & Carbin, 2019; Evci et al., 2019). Under specific settings, sparse networks have demonstrated superior performance than dense counterparts empirically, but the extent and the reason are still under investigation (Jin et al., 2022).

Many related methods, including this one, rely on the Straight-Through Estimator (STE) to define the gradient for quantization operators that generate the masks. Despite the lack of rigorous theoretical foundations (Shekhovtsov & Yanush, 2022), STE-based methods have demonstrated good empirical performance in training binarized or quantized neural networks (Hubara et al., 2016; Krishnamoorthi, 2018; Bethge et al., 2019; Alizadeh et al., 2019; Meng et al., 2020).

## 3 The BinMask method

### 3.1 General formulation

BinMask applies binary masks to the weights or inputs of neural networks. $L_0$ regularization is accomplished via regularizing the norm of binary masks. This formulation decouples the optimization of weight values from the optimization of masks, which allows recovering removed weights to their values before they are masked out (except changes by weight decay). Such a formulation is favorable because: (i) allowing the growth of connections has been shown to yield better sparse networks (Evci et al., 2020); and (ii) recovering weights to previous values is compatible with the finding that tuning pruned networks with earlier weight values or learning rate schedule is beneficial (Renda et al., 2020).

Formally, given a function $f : \mathbb{R}^n \mapsto \mathbb{R}^m$ representing the neural network with inputs and parameters packed as one argument, a loss functional $L : (\mathbb{R}^n \mapsto \mathbb{R}^m) \mapsto \mathbb{R}$ (i.e., a higher-order function that maps another function to a real value, which can include a training procedure of the neural network), an integer $k \in [1, n]$ denoting the number of values allowed to be masked out, and a penalty coefficient $\lambda \geq 0$, BinMask attempts to solve the following optimization problem:

$$\underset{\boldsymbol{b} \in \{0,1\}^k}{\arg\min} \ \tilde{L}(\boldsymbol{b}) \tag{1}$$
$$\text{where } \tilde{L}(\boldsymbol{b}) \overset{\text{def}}{=} L(f \circ g_{\boldsymbol{b}}) + \lambda \|\boldsymbol{b}\|_1$$
$$g_{\boldsymbol{b}}(\boldsymbol{x}) \overset{\text{def}}{=} \boldsymbol{x} \odot [\boldsymbol{b}; \boldsymbol{1}_{n-k}]$$

Assuming $L(\cdot)$ and $f(\cdot)$ can be evaluated in polynomial time, the corresponding decision problem of Eq. (1) (i.e., deciding if $\exists \boldsymbol{b} : \tilde{L}(\boldsymbol{b}) < c$) is NPC because it takes polynomial time to encode a 3-SAT instance in $L$ or verify a solution. Therefore, exactly solving Eq. (1) is NP-hard.

Assuming $\tilde{L}(\boldsymbol{b})$ is differentiable with respect to $\boldsymbol{b}$, we use a gradient-based optimizer to optimize real-valued latent weights $\boldsymbol{b}_r \in \mathbb{R}^k$ that are deterministically quantized to obtain the binary mask $\boldsymbol{b}$. We use the identity straight-through estimator to propagate gradient through the non-differentiable quantization operator, which has demonstrated good empirical performance in training binarized or quantized neural networks (Hubara et al., 2016; Krishnamoorthi, 2018). Formally, we transform Eq. (1) into the following problem:

$$\underset{\boldsymbol{b}_r \in \mathbb{R}^k}{\arg\min} \ \tilde{L}(q(\boldsymbol{b}_r)) \tag{2}$$
$$\text{where } q_i(\boldsymbol{b}) \overset{\text{def}}{=} \begin{cases} 1 & \text{if } \boldsymbol{b}_i \geq 0 \\ 0 & \text{if } \boldsymbol{b}_i < 0 \end{cases}$$

Eq. (2) is solved via a gradient-based optimizer. The identity straight-through estimator defines the gradient of $q(\cdot)$: $\frac{\partial \tilde{L}}{\partial \boldsymbol{b}_r} = \frac{\partial \tilde{L}}{\partial q(\boldsymbol{b}_r)}$.

The loss functional $L(\cdot)$ often includes an iterative optimization procedure of the network weights. We optimize $\boldsymbol{b}_r$ and neural network weights simultaneously in such cases. Since it has been suggested that during initialization, the weights undergo volatile changes (Leclerc & Madry, 2020), and pruning at initialization is more challenging (Frankle et al., 2021), we freeze the binary masks during initial epochs. We initialize the BinMask weights $\boldsymbol{b}_r$ with a constant value $\alpha_0$ and clip its values between $[-\alpha_1, \alpha_1]$ during training. To optimize $\boldsymbol{b}_r$, we use the Adam optimizer (Kingma & Ba, 2014), whose use of the second moment has been shown to be crucial for training performant binarized neural networks (Alizadeh et al., 2019). We use cosine learning rate annealing to schedule learning rates (Loshchilov & Hutter, 2017b). Since $\|\boldsymbol{b}\|_1$ is not differentiable at $\boldsymbol{b} = 0$ and $\lambda \|\boldsymbol{b}\|_1 = \frac{1}{2}\lambda \|\boldsymbol{b}\|_2^2$ for $\boldsymbol{b} \in \{0, 1\}^k$, we use the quadratic form in the implementation. Algorithm 1 summarizes our algorithm.

---

**Algorithm 1** Training with BinMask

---

**Input:** Dataset $D$
**Input:** Neural network function $f(\boldsymbol{x}, \boldsymbol{W})$ with initial weights $\boldsymbol{W}$ and loss function $L(\boldsymbol{y}, \hat{\boldsymbol{y}})$
**Input:** Total number of epochs $E$
**Input:** Number of iterations per epoch $T$
**Input:** An optimizer $U(\boldsymbol{W}, \nabla)$ for the neural network
**Input:** A specification $s$ describing to which weights and inputs should BinMask be applied
**Input:** BinMask regularization coefficient $\lambda$
**Input:** BinMask initialization $\alpha_0$ (default: 0.3)
**Input:** BinMask clip $\alpha_1$ (default: 1)
**Input:** BinMask initial learning rate $\eta_0$ (default: $10^{-3}$)
**Input:** BinMask final learning rate $\eta_1$ (default: $10^{-5}$)
**Input:** BinMask warmup epoch fraction $E_b$ (default: 0.1)
**Output:** Trained weights $\boldsymbol{W}^*$ and BinMask values $\boldsymbol{b}^* \in \{0, 1\}^{|s|}$.

1: $E_b' \leftarrow \lfloor E_b E \rceil$

2: $\boldsymbol{b}_r \leftarrow \alpha_0 \boldsymbol{1}_{|s|}$          ▷ Initialize $\boldsymbol{b}_r$ to be a constant vector corresponding to items described by $s$

3: **for** $e \leftarrow 0$ **to** $E - 1$ **do**

4:     **if** $e \geq E_b'$ **then**

5:        $\eta_b \leftarrow \eta_0 + \frac{\eta_1 - \eta_0}{2}\left(\cos\left(\frac{e - E_b'}{E - E_b' - 1}\pi\right) + 1\right)$          ▷ schedule BinMask learning rate

6:     **end if**

7:     **for** $t \leftarrow 0$ **to** $T - 1$ **do**

8:        Sample $(\boldsymbol{x}, \boldsymbol{y}) \sim D$

9:        $\boldsymbol{b} \leftarrow q(\boldsymbol{b}_r)$          ▷ $q(\cdot)$ defined in Eq. (2)

10:       $(\boldsymbol{x}', \boldsymbol{W}') \leftarrow s(\boldsymbol{x}, \boldsymbol{W}, \boldsymbol{b})$      ▷ Multiply binary masks with the weights or inputs specified by $s$

11:       $\boldsymbol{g}_W \leftarrow \frac{\partial}{\partial \boldsymbol{W}} L(f(\boldsymbol{x}', \boldsymbol{W}'), \boldsymbol{y})$

12:       $\boldsymbol{W} \leftarrow U(\boldsymbol{W}, \boldsymbol{g}_W)$          ▷ Update the weights with the user-provided optimizer

13:       **if** $e \geq E_b'$ **then**

14:          $\boldsymbol{g}_b \leftarrow \left(\frac{\partial}{\partial \boldsymbol{b}} L(f(\boldsymbol{x}', \boldsymbol{W}'), \boldsymbol{y})\right) + \lambda \boldsymbol{b}$

15:          $\boldsymbol{b}_r' \leftarrow \text{Adam}(\boldsymbol{b}_r, \boldsymbol{g}_b, \eta_b)$      ▷ Get updated BinMask weights with the Adam optimizer

16:          $\boldsymbol{b}_r \leftarrow \min\{\max\{\boldsymbol{b}_r', -\alpha_1\}, \alpha_1\}$

17:       **end if**

18:     **end for**

19: **end for**

20: **return** $\boldsymbol{W}^* \leftarrow \boldsymbol{W}$, $\boldsymbol{b}^* \leftarrow q(\boldsymbol{b}_r)$

---

### 3.2 Feature selection

We apply BinMask to the inputs of a network for feature selection. Note that if the first layer is fully connected, applying BinMask to the input is equivalent to applying BinMask to groups of the first layer weights, which is structural sparsification of the first layer by requiring that either all or none of the connections from each input to the neurons must be selected. Unfortunately, the BinMask algorithm described above is not ideal for practical feature selection for two reasons:

1. The binary mask may change after each iteration, making the result sensitive to the number of iterations.

2. Typical feature selection algorithms allow selecting a given number of features requested by the user rather than indirectly controlling the size of selected feature set through a regularization coefficient. However, it is challenging to control $\lambda$ to select a precise number of features with BinMask, especially given the randomness in minibatch sampling during training.

To ameliorate those two problems, we introduce a smoothed mask, denoted as $\hat{\boldsymbol{b}}$, for feature selection with BinMask. We compute the smoothed mask with moving average after line 9 in Algorithm 1:

$$\hat{\boldsymbol{b}} \leftarrow \gamma\hat{\boldsymbol{b}} + (1 - \gamma)\boldsymbol{b} \tag{3}$$

We initialize $\hat{\boldsymbol{b}}$ with zeros and set $\gamma = 0.9$ in our experiments. Let $\hat{\boldsymbol{b}}(\lambda)$ denote the smoothed mask computed with regularization coefficient $\lambda$. If the user provides a value of $\lambda$ without specifying the number of selected features, we select the $i^{\text{th}}$ feature if $\hat{\boldsymbol{b}}_i(\lambda) \geq 0.5$. Of note, most of the smoothed mask values converge to either zero or one. When the user requests exactly $k$ features, we need to find a $\lambda$ value to not arbitrarily cut features in the convergent region. We search for $\lambda^*$ and a threshold $c_{\lambda^*}$ such that $0.2 \leq c_{\lambda^*} \leq 0.8$ and $\left|\left\{i \mid \hat{\boldsymbol{b}}_i(\lambda^*) \geq c_{\lambda^*}\right\}\right| = k$. We use exponential search starting with $\lambda_0 = 10^{-3}$. We select the $i^{\text{th}}$ feature if $\hat{\boldsymbol{b}}_i(\lambda^*) \geq c_{\lambda^*}$.

## 4 Experiments

We evaluate the effectiveness of BinMask on three tasks: feature selection, network sparsification, and model regularization. We implement BinMask with a standalone Python module based on PyTorch (Paszke et al., 2019). Our implementation can be easily incorporated into other PyTorch projects and supports automatically patching most existing PyTorch networks to use BinMask for weight sparsification with minimum additional code. We will open source the implementation after paper review. We use the same implementation for all the experiments, with hyperparameters given in Algorithm 1 except for one change in feature selection.

### 4.1 Feature selection

**Datasets**: We evaluate the capability of BinMask and other methods to select a subset of features that maximize classification accuracy on five datasets. AP-Breast-Ovary is a binary classification dataset with gene expression features. BASEHOCK is a text classification dataset. Dexter is a text classification task with purposely added noninformative features used as a feature selection benchmark at NeurIPS 2003 (Guyon et al., 2004). Isolet is a speech classification dataset. MNIST is a handwritten digit classification dataset with pixel values as inputs. Note that for MNIST, feature selection is selecting a set of pixel locations to maintain classification accuracy. Our comparison methods also use BASEHOCK, Isolet, and MNIST. We select the other two datasets by querying OpenML (Vanschoren et al., 2013) for classification datasets with no missing features and at least 512 instances. We then manually check the top few datasets with the lowest instance-to-feature ratio to select the two datasets with a nonempty description. Features of each dataset are linearly normalized to the range $[0, 1]$. We use the provided training/test split of MNIST and randomly select 20% instances as the test set for other datasets.

**Classifiers**: For MNIST, we consider two neural network architectures. One is an MLP with two hidden layers, each with 512 neurons and ReLU followed by batch normalization (Ioffe & Szegedy, 2015), denoted as MNIST-MLP in the results. The other is a LeNet5 convolutional neural network (LeCun et al., 1998) with batch normalization after convolutional layers, denoted as MNIST-CNN. For other datasets, we use an MLP with two hidden layers, each having 64 and 20 neurons with tanh, respectively. MNIST networks are trained for 10 epochs and other networks for 100 epochs. For all tasks, we use the SGD optimizer with momentum 0.9 and weight decay $5\times10^{-4}$, with cosine learning rate annealing from 0.1 to $10^{-5}$. Each minibatch has 256 instances. If a dataset is too small, the training set is duplicated to have at least 30 minibatches per epoch.

**BinMask setup**: Since with the default setting of Algorithm 1 and the small training sets, the networks easily overfit the training data before any feature is masked out, making the gradients with respect to inputs

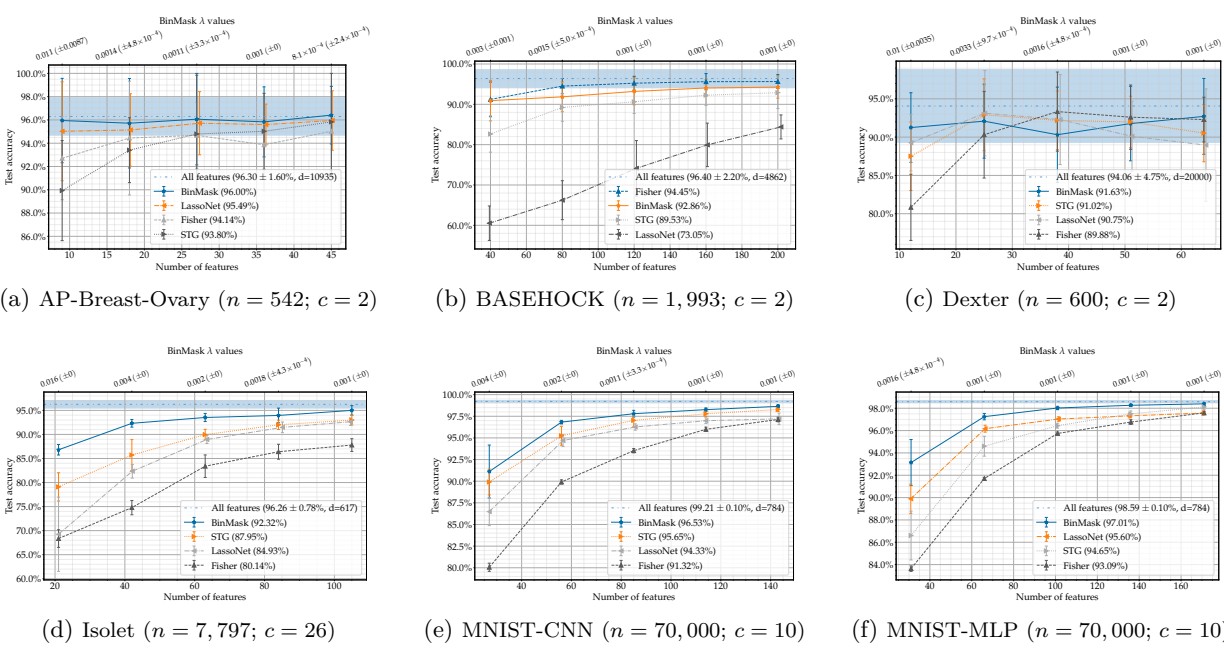

Figure 1: Feature selection results on various datasets. Bracketed numbers in the legends are mean accuracy over all numbers of features. The top horizontal axis, BinMask $\lambda$ values, are the values of regularization coefficients found by exponential search to select a given number of features, where bracketed numbers are standard deviation over eight trials. Error bars and the transparent horizontal region indicate 95% CI computed over eight trials. For each dataset, $n$ is the total number of data points, $c$ is the number of classes, and $d$ is the number of input features.

uninformative for mask optimization, we set the initialization $\alpha_0 = 0.02$ to allow easier mask flip before overfitting. Other hyperparameters use their default values. For each task, we apply BinMask to the inputs of a network with the same architecture as the classification network for that task.

**Comparison**: We compare BinMask with three other methods. Fisher score is a classical filter model for supervised feature selection that aims to find a subset of features to maximize interclass distance while minimizing intraclass distance (Gu et al., 2011). We use the implementation provided by `scikit-feature` (Li et al., 2018). Fisher score produces a weight for each feature that can be ranked to select a given number of features. Stochastic gates (STG) uses a probabilistic relaxation to regularize the $L_0$ "norm" of features while simultaneously learning a classifier (Yamada et al., 2020). We use the open-source implementation provided by the authors[1]. For each task, we apply the STG layer to the inputs of a network with the same architecture as the classification network for that task. We use their default hyperparameter settings. STG produces probabilities for selecting each input feature. Since the STG paper does not specify an algorithm to choose a probability cutoff threshold for feature selection, we use the same exponential search strategy outlined in Section 3.2 to find a suitable regularization coefficient for STG, starting with the default value 0.1 given by their implementation. LassoNet uses a neural network with skip connections and $L_1$ regularization to simultaneously learn linear and nonlinear components for the task (Lemhadri et al., 2021). We use the open-source implementation provided by the authors[2]. We use their default hyperparameter settings and a one-hidden-layer neural network, as suggested by the authors. LassoNet produces a path of networks with decreasing numbers of features used. We choose from the path the feature set whose size is closest to the given number of features.

---

[1] https://github.com/runopti/stg
[2] https://github.com/lasso-net/lassonet

**Evaluation**: For a task $t$, we first use BinMask with $\lambda = 0.001$ ($\lambda = 5 \times 10^{-4}$ for AP-Breast-Ovary) without feature number constraint to select a set of features. Let $n_t$ be the number of features selected. We then run all methods to select $n_t - i\lfloor n_t/5 \rfloor$ features for $0 \leq i \leq 4$. Given a set of selected features by a method, we evaluate the accuracy of the classifier network corresponding to the task trained from new random initialization with selected features. We only keep selected features as network inputs, except for MNIST-CNN where we use zeros to replace unselected features so that convolutional layers receive two-dimensional inputs. We run eight trials with different dataset split and different weight initialization for each (dataset, method, number of features) configuration. We evaluate each method by its mean accuracy on the test set over eight trials. Since BinMask (with smoothed masks) and STG (Yamada et al., 2020) produce continuously valued masks that should converge to zero or one in the ideal situation, we also evaluate how likely the computed masks converge. A mask is defined to have converged if at most 20% of its values are within $[0.15, 0.85]$.

**Results**: Fig. 1 presents the accuracies on the considered benchmarks, showing that BinMask, a generic $L_0$ regularizer, exhibits competitive performance compared to other methods tailored to feature selection. BinMask achieves best average accuracies on five of the six benchmarks. STG (Yamada et al., 2020) uses 1.6 steps for the exponential search on average, and 8.86% of the computed masks converge. By comparison, BinMask uses 1.7 search steps, with 75.00% of the masks converging.

## 4.2 Network sparsification

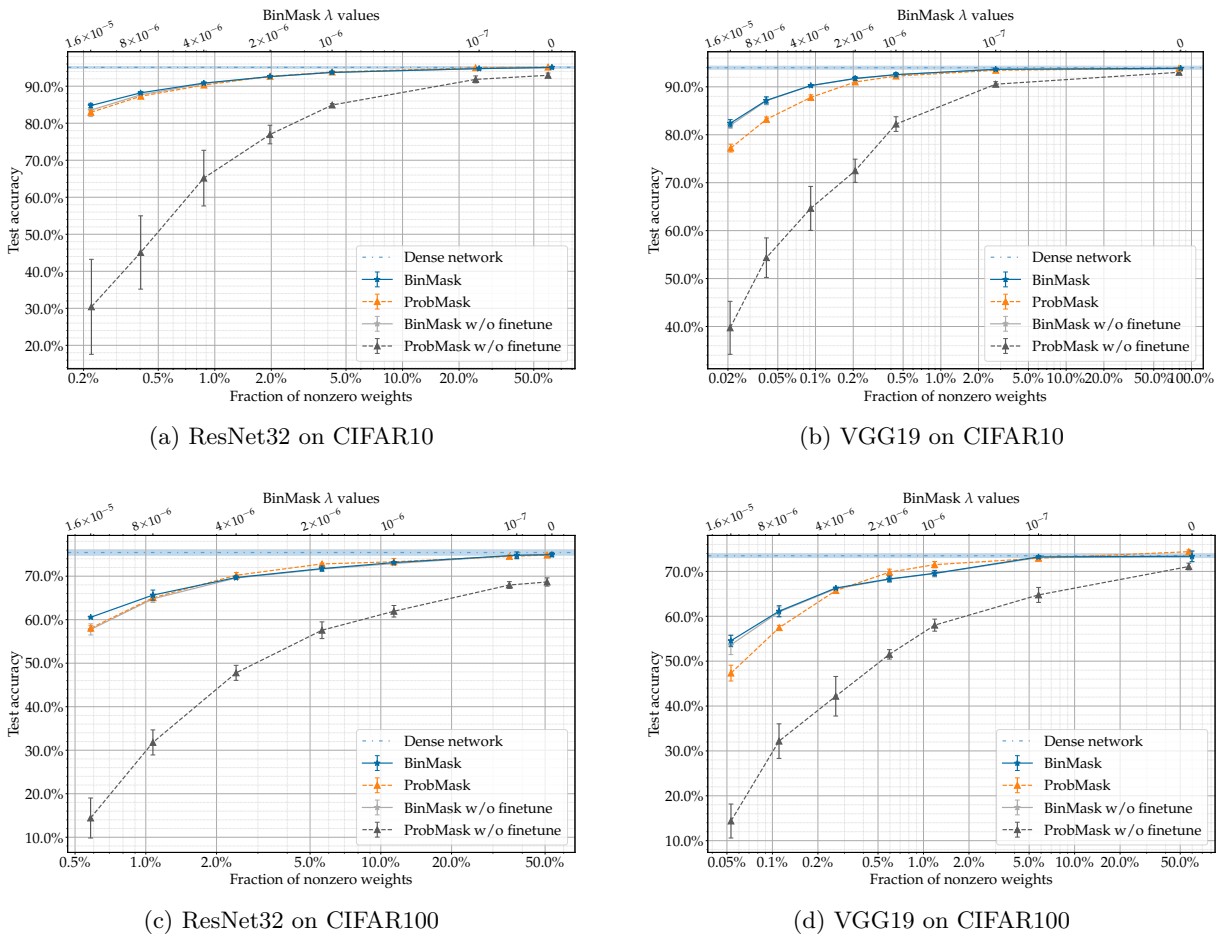

(a) ResNet32 on CIFAR10

(b) VGG19 on CIFAR10

(c) ResNet32 on CIFAR100

(d) VGG19 on CIFAR100

Figure 2: Network sparsification results on CIFAR10/CIFAR100. Error bars and the transparent horizontal region indicate 95% CI computed over four trials.

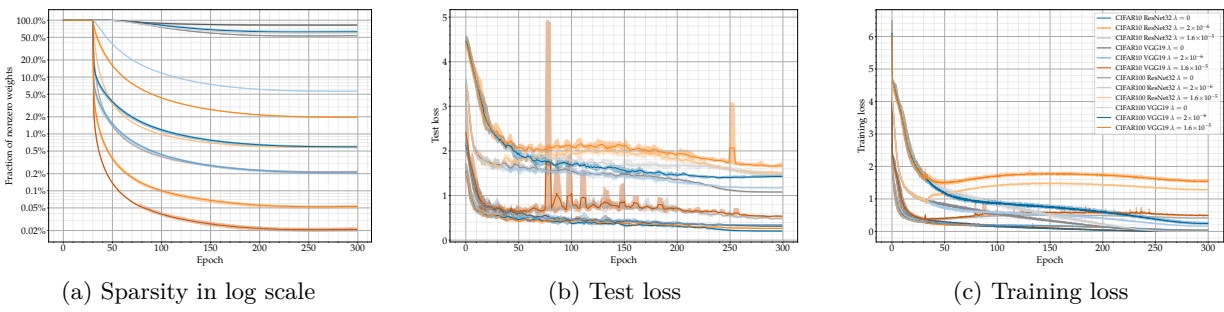

(a) Sparsity in log scale  (b) Test loss  (c) Training loss

Figure 3: Metrics at each training step for selected BinMask $\lambda$ values. Filled regions of each line are min/max values over four trials.

We apply BinMask to train sparse ResNet32 (He et al., 2016) and VGG19 (Simonyan & Zisserman, 2014) networks on CIFAR-10/100 datasets (Krizhevsky et al., 2009). We sparsify the weights (excluding biases) of convolutional and fully connected layers. We compare with ProbMask (Zhou et al., 2021), a state-of-the-art method for training sparse networks with a probabilistic relaxation to $L_0$ constraint. We their authors' open-source implementation[3]. ProbMask gradually sparsifies the network during training to meet a predefined target sparsity. We use the same setting as ProbMask for weight optimization: SGD optimizer for 300 epochs with momentum of 0.9 and cosine learning rate annealing starting from 0.1. The ProbMask implementation finetunes the model for 20 epochs after training the masks. For a fair comparison, we also conduct finetuning for the dense models and BinMask models with fixed masks and report results before and after finetuning.

For each network architecture and dataset, we train BinMask networks with $\lambda \in \{1.6 \times 10^{-5}, 8 \times 10^{-6}, 4 \times 10^{-6}, 2 \times 10^{-6}, 10^{-6}, 10^{-7}, 0\}$. We then train ProbMask networks with the same overall sparsity as BinMask networks. For each configuration, we run four trials and evaluate the mean accuracy on the test set.

Fig. 2 presents the test accuracies. One can immediately notice the significant performance drop experienced by ProbMask without finetuning. A primary reason is that resampling the probabilistic masks causes shift in the batch normalization statistics. Using per-minibatch statistics instead of the running mean/variance in batch normalization improves ProbMask's mean test accuracy by 12.9% to 17.8% in the four configurations. By contrast, BinMask uses deterministic masks and does not exhibit a large performance drop without finetuning. Notably, with $\lambda \in \{0, 10^{-7}\}$, BinMask produces sparse networks with up to 97% sparsity without statistically significant performance differences from the dense networks. BinMask delivers worse accuracies than ProbMask in a few CIFAR-100 configurations but constantly outperforms ProbMask in all high-sparsity cases. BinMask is computationally efficient. On a single NVIDIA Titan XP GPU, training with BinMask is 3.40% slower for ResNet32 and 10.97% slower for VGG19 than training the dense networks. By comparison, ProbMask is 1.4x and 1.9x slower than BinMask on ResNet32 and VGG19, respectively.

Fig. 3 presents the change of sparsity, test loss, and training loss during training. It shows that BinMask can improve sparsity quickly during training without notably impacting training or test loss.

### 4.3 Model regularization

The above experiments have not shown any advantage of sparse networks regarding predictive power. This section evaluates BinMask's capability to regularize neural networks for better generalizability. We work with a dataset for early prediction of Pancreatic Duct Adenocarcinoma (PDAC) diagnosis, a type of pancreatic cancer, based on Electronic Health Records (EHR) (Jia et al., 2023). The PDAC dataset contains features extracted from EHR of millions of patients. The features are sparse representations of diagnosis, medication, and lab results before a cutoff date for each patient. The outcome variable is PDAC diagnosis within 6 to 18 months after the cutoff date. The PDAC dataset has 63,884 positive instances and 3,604,863 negative instances, with over five thousand features per instance (the exact number of features vary slightly depending on training/test split because the EHR codes for feature generation are selected based on a frequency

---

[3]https://github.com/x-zho14/ProbMask-official

Table 1: Network regularization results on the PDAC dataset.

| Metric | Method | Values (95% CI) | | | | |
|---|---|---|---|---|---|---|
| Regularization coeff. / dropout prob. | BinMask | $3\times10^{-5}$ | $4\times10^{-5}$ | $2\times10^{-5}$ | $5\times10^{-5}$ | $10^{-5}$ |
| | $L_1$ | $10^{-4}$ | $5\times10^{-5}$ | $2\times10^{-4}$ | $3\times10^{-4}$ | $10^{-5}$ |
| | $L_2$ | 0.01 | 0.1 | 10 | 1 | 50 |
| | Dropout | 0.5 | 0.3 | 0.7 | 0.1 | 0.9 |
| Training AUC | BinMask | 0.858(±0.003) | 0.850(±0.003) | 0.874(±0.003) | 0.844(±0.007) | 0.907(±0.020) |
| | $L_1$ | 0.859(±0.004) | 0.880(±0.008) | 0.846(±0.005) | 0.838(±0.003) | 0.892(±0.021) |
| | $L_2$ | 0.879(±0.028) | 0.890(±0.038) | 0.831(±0.006) | 0.914(±0.041) | 0.796(±0.004) |
| | Dropout | 0.908(±0.003) | **0.942(±0.024)** | 0.862(±0.002) | 0.918(±0.043) | 0.804(±0.004) |
| | LR | 0.834(±0.004) | | | | |
| Test AUC | BinMask | **0.834(±0.006)** | 0.832(±0.007) | 0.830(±0.007) | 0.830(±0.010) | 0.815(±0.009) |
| | $L_1$ | **0.836(±0.007)** | 0.832(±0.007) | 0.831(±0.007) | 0.828(±0.007) | 0.823(±0.010) |
| | $L_2$ | 0.820(±0.008) | 0.819(±0.009) | 0.816(±0.007) | 0.814(±0.017) | 0.793(±0.008) |
| | Dropout | 0.827(±0.006) | 0.824(±0.008) | 0.824(±0.006) | 0.820(±0.012) | 0.798(±0.007) |
| | LR | 0.819(±0.008) | | | | |
| Mean weight $L_0$ | BinMask | 0.018(±0.001) | 0.013(±0.001) | 0.027(±0.001) | 0.010(±0.001) | 0.178(±0.277) |
| | $L_1$ | 0.071(±0.006) | 0.161(±0.073) | 0.044(±0.006) | 0.037(±0.002) | 0.966(±0.007) |
| | $L_2$ | 0.999(±0.000) | 0.999(±0.000) | 0.967(±0.002) | 0.995(±0.001) | 0.960(±0.004) |
| | Dropout | 0.999(±0.000) | 0.999(±0.000) | 0.999(±0.000) | 0.999(±0.000) | 0.999(±0.000) |
| Mean weight $L_1$ | BinMask | 0.005(±0.000) | 0.003(±0.000) | 0.007(±0.000) | 0.003(±0.000) | 0.020(±0.009) |
| | $L_1$ | 0.001(±0.000) | 0.002(±0.000) | 0.000(±0.000) | 0.000(±0.000) | 0.017(±0.001) |
| | $L_2$ | 0.051(±0.009) | 0.056(±0.011) | 0.002(±0.000) | 0.017(±0.003) | 0.002(±0.000) |
| | Dropout | 0.118(±0.001) | 0.134(±0.007) | 0.103(±0.001) | 0.096(±0.020) | 0.088(±0.001) |
| Mean weight $L_2$ | BinMask | 0.044(±0.001) | 0.038(±0.001) | 0.055(±0.001) | 0.034(±0.001) | 0.077(±0.006) |
| | $L_1$ | 0.009(±0.000) | 0.014(±0.000) | 0.006(±0.001) | 0.005(±0.000) | 0.039(±0.003) |
| | $L_2$ | 0.068(±0.012) | 0.073(±0.014) | 0.004(±0.000) | 0.023(±0.003) | 0.003(±0.000) |
| | Dropout | 0.156(±0.001) | 0.176(±0.009) | 0.136(±0.001) | 0.125(±0.026) | 0.115(±0.001) |

Notes:
- Columns are sorted by the test AUCs of each method.
- Bracketed numbers are 95% CI computed over eight trials.
- LR is logistic regression.
- Mean weight $L_0$ is the fraction of weights whose absolute value is at least $10^{-4}$.
- Let $\boldsymbol{W} \in \mathbb{R}^n$ be the flattened weight. For $p \geq 1$, mean weight $L_p$ is $\left(\frac{1}{n}\sum_{1\leq i\leq n}|\boldsymbol{W}_i|^p\right)^{\frac{1}{p}}$.
- **Bold** numbers are AUCs than which the best training/test AUC is not significantly better ($p > 0.3$).

threshold on the training set). On average, each feature has a 94% chance of being zero. The dataset can be obtained under some collaboration agreement (details to be revealed after paper review). The dataset is split into training, validation, and test sets, with 75%, 10%, and 15% of the instances, respectively.

We train MLP neural networks with two hidden layers, each having 64 and 20 neurons with tanh, respectively. We use the AdamW (Loshchilov & Hutter, 2017a) optimizer and cosine learning rate annealing from 0.002 to $5\times10^{-5}$ with 16 epochs. We consider four regularization methods: $L_0$ regularization with BinMask, $L_1$ regularization (a.k.a., Lasso regularization), $L_2$ regularization (a.k.a., weight decay), and dropout (Srivastava et al., 2014). We test five regularization coefficients (or dropout probabilities) for each method. For methods other than $L_2$ regularization, we also use a weight decay of 0.01. We adopt early stopping for all methods by selecting the model with the highest validation AUC after each epoch.

We evaluate model performance by computing their mean AUC on the test sets over eight trials with different dataset split and weight initialization. We also train Logistic Regression (LR) models for reference. Table 1 presents the results. Our results show that all the considered regularization methods can effectively regularize the corresponding weight norm, but only $L_0$ regularization with BinMask and $L_1$ regularization significantly improve network generalizability. While it has been suggested that in most clinical cases, NN provides marginal to no improvement over LR (Issitt et al., 2022; Appelbaum et al., 2021), our results suggest that with proper regularization, NN can achieve significant improvement in clinical applications.

## 5 Conclusion

This paper revisits a straightforward formulation, BinMask, for $L_0$ regularization of neural networks. The critical insight is decoupled optimization of weights and binary masks. The binary mask is optimized via the identity straight-through estimator and the Adam optimizer, which are widely accepted choices for training binarized neural networks. We demonstrate competitive performance of BinMask on diverse benchmarks, including feature selection, network sparsification, and model regularization, without task-specific hyperparameter tuning.

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
