# OpenReview forum: "Effective Neural Network $L_0$ Regularization With BinMask"
_TMLR — Rejected by TMLR_

### Review · Reviewer_W7Nj · 2023-05-12

**Summary Of Contributions:**

This work revisits BinMask, a method previously proposed for learning sparse binary neural networks, in the context of learning sparse neural networks with the $L_0$ norm and real valued non-zero elements. The method is quite simple and it works by binarizing real values parameters to get sparse masks and then using them to multiply the real valued weights of the neural network. To optimise the non-differentiable binary masks, the authors use the popular straight-through estimator (STE), i.e., treating the quantization operations as an identity function in the backward pass. The authors further discuss some special considerations one needs for BinMask when employing it for feature selection, namely, a moving average for the mask in order to smooth it out and be able to tune a threshold that selects a specific number (user defined) of features. The authors do experiments on three settings, feature selection, network sparsification and model regularisation, showing improvements upon the baselines considered.

**Audience:**

No

**Broader Impact Concerns:**

No concerns.

**Claims And Evidence:**

No

**Requested Changes:**

I am leaning a bit on the negative side for this work; the main idea, BinMask, has been proposed before and the experimental settings the authors explore are not broad enough to provide reasonable insights. Therefore, my main requests / points for discussion are the following

- Increase scale of the experiments as well as the baseline methods. For example, the authors could include experiments on ImageNet with ResNet50 / MobileNet (as, e.g., done at RigL [1]) for the model sparsification settings, along with additional baselines there, e.g. [1],[2].
- Methods such as [2] seem more natural for the feature selection task considered, since they allow for direct optimization for the number of features (instead of implicitly via the regularisation strength). I believe the authors should include such methods in this specific comparison
- The model regularisation experiments are not really informative; most methods seem to be within the (I assume to be) standard deviation, so it is kind of hard to argue for one over the other. I would also have expected more datasets in this setting, to improve the generality of the conclusions. Also, why are methods such as ProbMask not considered in this setting? The stochasticity might be beneficial for regularisation purposes.
- The authors mention that the BinMask binary masks may change after each iteration in the feature selection task. This is kind of peculiar, as the authors also argue that the masks properly converge (e.g., discussion of the results above section 4.2). Which of the two apply? It would be interesting if the authors have a plot that show, e.g., the $L_1$ norm of the difference of the masks (or probabilities in the case of ProbMask) between subsequent iterations over time to see exactly whether the method properly converges.
- The notion of the smooth mask is a bit weird; if one needs a smooth mask they why not train with methods that provide those smooth masks directly, e.g., ProbMask? Furthermore, do I understand correctly that after a specific set of features is selected, the neural network is trained from scratch on those features only?
- For the feature selection task, is the same $\alpha_0 = 0.02$ initialisation applied to STG as well?
- One of the main explanations of the worse performance of ProbMask at Figure 2 is that the sampling causes shifts in the batch norm statistics. While this could be a problem in the beginning of training, the authors of ProbMask show that the mask probabilities converge to deterministic values at later stages of training, so the shift in the batch normalisation statistics shouldn’t be an issue. What do the authors observe in their experiments and have they trained long enough to allow for proper convergence?
- One of the claims is that "decoupling weights from mask optimization is a key component for effective $L_0$ regularization". Which experiment demonstrates that insight? I would have expected experiments that demonstrate how methods that couple them result into worse performance compared to those who don't.

Miscellaneous things
- The visibility of most of the plots is a bit poor. I encourage the authors to increase the line widths and also the font sizes for the legends / axis descriptions. Furthermore, I encourage the authors to have consistent colouring of the methods; for example in Figure 1 BinMask is blue in 1(a) but orange in 1(b).
- Notation at eq. 1 is a bit peculiar; is $\mathbf{x}$ the concatenation of both inputs to the network and parameters?
- The first paragraph of section 3.1 is a bit unclear; what do the authors mean with “recovering removed weights to their values before they are masked out”?
- The authors should also add ProbMask sparsity rates on Figure 3(a).

**Strengths And Weaknesses:**

Strengths
- Method is quite simple and straightforward
- There is no randomness in the masking selection procedure, which can facilitate for easier optimization
- Paper is relatively well written and easy to follow

Weaknesses
- As the main contributions of this work are on the experimental side, I would have liked larger scale experiments, e.g., ImageNet
- Experiments are also missing important baselines; for example magnitude-based pruning, e.g., RigL [1], and constrained based optimization for the L0 norm [2], to, e.g., allow for natural selection of the number of features in the feature selection experiment.

[1] Rigging the lottery: making all tickets winners, Evci et al., 2020

[2] Controlled Sparsity via Constrained Optimization, Gallego-Posada et al., 2022

---

### Review · Reviewer_amme · 2023-05-17

**Summary Of Contributions:**

The paper the BinMask method for sparse feature selection and network pruning (sparse weights).

**Audience:**

Yes

**Claims And Evidence:**

No

**Requested Changes:**

Please see above for changes.

**Strengths And Weaknesses:**

### Novelty and Correctness
I am not familiar with the historical development of sparsification techniques based on straight-through estimator (STE). However, the authors appear to revisit a known technique, BinMask, for sparsification. I cannot comment on the novelty of the approach and leave the discussion to the other reviewers.

In terms of correctness, there are several issues that need to be addressed:
- The description of the problem (1) needs to be improved. First, using a loss "functional" obfuscates the representation and hides the fact that 1) the problem also depends on training data, 2) the optimization is jointly over the network parameters and the mask. Using a simple loss function for the problem description is much more standard and cleaner. Also, the ordered concatenation of the mask with an all-one variable of length $(n-k)$ makes it look like the sparsity is only applied to the first $k$ elements (which I assume is not true).
- The optimization problem (1) applies L1 norm penalty. However, the authors replace L1 with a differentiable squared L2 norm regularizer based on the fact that for a binary mask, the values of the regularizers are the same. However, the values of the two regularizers are different non-binary vectors, which are, in fact, used in the optimization. I do not see an issue with the non-differentiability of L1, as autodiff packages can handle the non-smooth regions.
- The NP-hardness of the problem is claimed without rigorous proof.
- The thresholding step at the end of Section 3 has no clear justification; why is the threshold value set to 0.5? The mask values do not have a probabilistic interpretation, and the value of 0.5 for the threshold seems rather arbitrary. Also, I did not quite follow the reason for an "exponential search" for finding a threshold to induce $k$ non-zero elements. Why not simply keep the top-$k$ largest absolute values of $\mathbf{b}$?
- Why does EMA fix the issue of changing sparsity patterns? The sparsity variable is continuous and thresholded to make it a binary mask. To control rapid changes in the sparsity variable, you can also use a regularizer or a lower learning rate. Also, the second point in Section 3.2 is irrelevant for justifying using EMA: how does EMA fix the issue with thresholding?

## Writing and Presentation
First, the paper requires a major revision to improve the writing. In the Related Work section, there are multiple ambiguous one-line descriptions of previous approaches: "they do not evaluate BinMask on other applications" (What are other applications?), "Louizos et al. (2018) use one sample per minibatch despite larger variance" (Variance in what exactly?), "preserve continuous gate variables... which is different from how selected features are used...thus causes lower performance." (What exactly causes lower performance, and lower compared to what?)...

Algorithm 1 has 12 input arguments in separate lines, most of which seem redundant details for the description of the main approach. Similarly, the exact description of what a cosine schedule is seems rather unnecessary. Also, the notation is again unclear at several spots: What does $E_b E$ with brackets mean? By sample $(x, y) \sim D$, do you mean sample a mini-batch? $x$ in (1) refers to the network parameters, but now $x$ is the input, and $W$ are the weights, etc.

---

### Review · Reviewer_Y31k · 2023-06-03

**Summary Of Contributions:**

The paper revisits an existing approach, BinMask (Jia & Rinard, 2020), for L0 regularization. The method is based on multiplying the weights with deterministic binary masks, which are jointly optimized. This paper evaluates BinMask on three different tasks: feature selection, network sparsification, and model regularization, and shows comparable or improved test-accuracy than several baseline methods.


**Audience:**

Yes

**Claims And Evidence:**

No

**Requested Changes:**

Please see weaknesses above that need to be addressed.

The formulation of the problem in eq (1) is non-standard and unnecessarily confusing. It'd be good to make clear what the symbols mean in eq (1), specifically $x$ represents the input/NN weights, $\circ$ denotes function decomposition.




**Strengths And Weaknesses:**

Strengths

The paper is fairly well presented and easy to follow. The method is simple to implement and has few hyperparameters.

Weaknesses

- It's unclear exactly what the methodological innovations are over the existing BinMask method proposed in (Jia & Rinard, 2020). The paper seems to mostly be about additional experimental evaluation of a specific method, which I'm not sure is a significant enough contribution of interest to the wider community.

- Even the experiments section do not seem very convincing.
  1. As ProbMask is a generally applicable L0 regularization method, why is it only compared against in Sec 4.2, but not the other experiments.
  2. Similarly, the hard-concrete method by (Louizos et al., 2018) is well known and should also be added as a baseline in the experiments. The method was dismissed in Related Work for having high-variance gradient estimates, but there's no comparison with the BinMask approach showing the latter using STE has lower-variance gradient. In my experience using hard-concrete (Louizos et al., 2018) this has not been an issue.
  3.  The experiment on a single dataset in Sec 4.3 does not convincingly justify claims about better generalization from BinMask. Compare this to the dropout paper (Srivastava et al., 2014), which evaluates on supervised learning tasks in vision, speech recognition, document classification and computational biology.

- The feature selection procedure in Sec 3.2 seems rather ad-hoc. It's noted that BinMask has the issue #1 that "the binary mask may change after each iteration, making the result sensitive to the number of iterations" -- does this indicate convergence issues for the binary masks in general? If no, why is it specific to weights in the first (input) layer? If yes, wouldn't be a good idea to apply the moving average (eq 3) for all the masks?  It'd be good to have a more principled development along with justifications and empirical evaluations of the design choices. As for the issue #2 (targeting an exact target of number of features), there are standard constrained optimization techniques such as the augmented Lagrangian method or modified method of multipliers (Platt and Barr, NIPS 1988) that should be compared with.

---

### Author Response · Authors · 2023-06-05
**Response to reviews**

We thank all the reviewers for the careful reviews and constructive feedback. Here is our response:

A major concern is the novelty of the method. The message we'd like to deliver through our paper is that for $L_0$ regularization of neural networks, a simple method, BinMask, has been overlooked. It achieves good results but is not compared with in prior work. Therefore, we believe using the simplest formulation without any unnecessary complication is a favorable choice. We hope to keep this paper short and easy to follow, with accessible open-source implementation for practitioners.

We compared only with ProbMask for network sparsification because it is state of the art. We compared with ProbMask using the code provided by the authors on their chosen datasets and network architectures, which has presumably been well-tuned for those tasks. By demonstrating comparable or even better performance than ProbMask (and thus better than other methods that ProbMask has compared with), we have shown that BinMask is a practical and useful method.

In our method presentation, we agree that our formulation (Eq. 1) is nonstandard. Although we believe our current formulation is the most general and high-level way to describe a method for $L_0$ regularization (which can be applied to weights, inputs, groups of weights, and potentially other structures), we are happy to rewrite this part of the paper to make it easier to understand and better conform to the tradition of this field.

Our feature selection method appears ad-hoc, but it also shows that BinMask is useful --- by wrapping an ad-hoc layer around it, we can do feature selection. Feature selection produces a given number of features. Although the convergence of BinMask is good enough for network sparsification, in the feature selection evaluation framework, the change of a single mask bit (which can result from one more iteration or different initialization) would make it difficult to select a given number of features. Therefore, we introduced the smoothed mask to solve the practical challenge of needing a specific number of features.

Below are responses to some individual points:

> The experiment on a single dataset in Sec 4.3 does not convincingly justify claims about better generalization from BinMask.

The generalization power of BinMask in Sec 4.3 comes from $L_0$ regularization. It is not necessarily better than other regularization techniques. This section aims to show that BinMask delivers effective $L_0$ regularization, which is not well studied in prior work since $L_0$ regularization is not considered a standard regularization technique for better generalizability for deep neural networks.

> I do not see an issue with the non-differentiability of L1, as autodiff packages can handle the non-smooth regions.

Using $L_0$, $L_1$, or $L_2$ in our formulation of (eq 1) results in equivalent optimization objectives. In practice, gradients of $L_1$ at zero can potentially be mishandled by autodiff packages. See, for example, [issue 7172](https://github.com/pytorch/pytorch/issues/7172) of pytorch.

> The NP-hardness of the problem is claimed without rigorous proof.

We are happy to extend the paragraph into a formal proof.

> Why not simply keep the top-$k$ largest absolute values of $\hat{b}$?

As noted in the paper, most values converge to zero or one. Without searching for a regularization coefficient, the top-$k$ values can be impacted by random factors (e.g., selecting a feature $\hat{b_i}=0.99998$ but not $\hat{b_j}=0.99997$).

> Furthermore, do I understand correctly that after a specific set of features is selected, the neural network is trained from scratch on those features only?

That's correct, as mentioned at the top of page 7 in the "Evaluation" paragraph.

> For the feature selection task, is the same initialisation $\alpha=0.02$ applied to STG as well?

No. We used author-recommended settings with their code, since some of the benchmarks are also considered in their paper.

> What do the authors observe in their experiments and have they trained long enough to allow for proper convergence?

As mentioned earlier, we used the official code of ProbMask, and trained for 300 epochs as written in their configuration file.

> I would have expected experiments that demonstrate how methods that couple them result into worse performance compared to those who don't

Many papers share this type of formulation. As we achieved competitive results using a simple implementation under this formulation, while other papers using this type of formulation also report good results, our claim is supported. It is also unclear what a fair comparison looks like (how to keep other parts unchanged while using a coupled formulation).

---

### Decision · Action_Editors · 2023-06-27

**Recommendation:** Reject

**Comment:**

This work considers a method for learning sparse binary neural networks. The method involves binarizing network parameters by quantizing them using the straight-through estimator. The authors propose a moving average approach for mask smoothing and threshold tuning and evaluate their method based on feature selection, network sparsification, and model regularization tasks.

The reviewers appreciated the simplicity of the approach, and the fact that no randomness is involved in the mask selection procedure. Unfortunately, the paper has a number of weaknesses that were not sufficiently addressed in the rebuttal period.

The paper did not present methodological contributions beyond the existing BinMask method since the paper mainly focused on additional experimental evaluations. There were furthermore several open questions on the experimental design, the feature selection procedure, the concept of smooth masks, and a lack of comparisons with other well-known baseline methods. The reviewers recommended extending the discussion on aspects such as regularizer choices, thresholding steps, proofs of NP-hardness, and additional experiments on larger datasets.

**Audience:**

Yes.

**Claims And Evidence:**

No. The paper did not present methodological contributions beyond the existing BinMask method since the paper mainly focused on additional experimental evaluations. There were furthermore several open questions on the experimental design, the feature selection procedure, the concept of smooth masks, and a lack of comparisons with other well-known baseline methods. Reviewer concerns were not sufficiently addressed. See below for details.